# Light-induced depigmentation in planarians models the pathophysiology of acute porphyrias

Bradford M Stubenhaus[1], John P Dustin[1], Emily R Neverett[1], Megan S Beaudry[1], Leanna E Nadeau[1], Ethan Burk-McCoy[1], Xinwen He[2,3], Bret J Pearson[2,3,4], Jason Pellettieri[1]*

[1]Department of Biology, Keene State College, Keene, United States; [2]The Hospital for Sick Children, Toronto, Canada; [3]Department of Molecular Genetics, University of Toronto, Toronto, Canada; [4]Ontario Institute for Cancer Research, Toronto, Canada

**Abstract** Porphyrias are disorders of heme metabolism frequently characterized by extreme photosensitivity. This symptom results from accumulation of porphyrins, tetrapyrrole intermediates in heme biosynthesis that generate reactive oxygen species when exposed to light, in the skin of affected individuals. Here we report that in addition to producing an ommochrome body pigment, the planarian flatworm *Schmidtea mediterranea* generates porphyrins in its subepithelial pigment cells under physiological conditions, and that this leads to pigment cell loss when animals are exposed to intense visible light. Remarkably, porphyrin biosynthesis and light-induced depigmentation are enhanced by starvation, recapitulating a common feature of some porphyrias – decreased nutrient intake precipitates an acute manifestation of the disease. Our results establish planarians as an experimentally tractable animal model for research into the pathophysiology of acute porphyrias, and potentially for the identification of novel pharmacological interventions capable of alleviating porphyrin-mediated photosensitivity or decoupling dieting and fasting from disease pathogenesis.

*For correspondence:
jpellettieri@keene.edu

**Competing interests:** The authors declare that no competing interests exist.

## Introduction

The prosthetic group heme plays a key role in proteins (hemoproteins) with diverse cellular functions, including oxygen binding (hemoglobin and myoglobin), electron transfer (cytochromes), protection from oxidative stress (catalases and peroxidases), and generation of second messengers (nitric oxide synthase and guanylate cyclase) (*Ponka, 1999*). With few exceptions (*Kořený et al., 2012*), it is essential for viability in all 3 domains of life.

Heme is composed of a heterocyclic tetrapyrrole called a porphyrin coordinated to a central iron atom (some researchers classify the entire heme molecule as a porphyrin, but this term is reserved for the organic ring alone here). Although naturally occurring auxotrophs have been reported (*Rao et al., 2005*; *Gruss et al., 2012*), heme is usually synthesized from the precursor 5-aminolevulinic acid (ALA) through a series of enzyme-catalyzed reactions. In non-photosynthetic eukaryotes, ALA is produced in the mitochondria via condensation of glycine and succinyl CoA by ALA synthase (ALAS). This is the first and typically rate-limiting step in the heme biosynthesis pathway, which proceeds in the cytoplasm before returning to the mitochondria (*Layer et al., 2010*). Like ALAS, the enzymes catalyzing the 7 remaining reactions in the pathway are tightly regulated by ubiquitous and tissue-specific mechanisms (*Ponka, 1997*; *Balwani and Desnick, 2012*).

**eLife digest** Porphyrias are rare diseases that involve ring-shaped molecules called porphyrins accumulating in various parts of the body. Porphyrins are produced as part of the normal process that makes an important molecule called heme, which is required to transport oxygen. However, high levels of porphyrins can be toxic. For example, porphyrins deposited in the skin can cause swelling and blistering when the skin is exposed to bright light. Other disease symptoms include neurological issues ranging from anxiety and confusion to seizures or paralysis. It has been speculated that porphyrias may have affected several historical figures, including the artist Vincent van Gogh.

In addition to their role in heme production, porphyrins also have other roles. For example, they are used as pigments in the wing feathers of some owls. Researchers are trying to understand more about how organisms regulate porphyrin production so that it might be possible to develop more effective treatments for porphyria in humans.

Here, Stubenhaus et al. studied how a flatworm called *Schmidtea mediterranea* makes porphyrins. A group of undergraduate students noticed that these animals – which are normally brown in color – turned white when they were exposed to sunlight for several days. Stubenhaus et al. found that *S. mediterranea* makes porphyrins in the pigment cells of its skin using the same genes that make porphyrins in humans. Together with other molecules called ommochromes, the porphyrins give rise to the normal color of this flatworm. However, when the animals are exposed to intense light for extended periods of time, which is unlikely to occur in the wild, porphyrin production leads to loss of the pigment cells.

The experiments also show that starvation increases the rate of pigment cell loss in light-exposed flatworms, which mirrors the worsening of disease symptoms some porphyria patients experience when they diet or fast. Stubenhaus et al. propose that flatworms are useful models in which to study the molecular processes that are responsible for porphyrias in humans. Further research is required to determine the exact chemical structure of the porphyrin and ommochrome molecules produced in different flatworm species. Stubenhaus et al. also plan to use flatworms to screen for drugs that could potentially be developed into new treatments for porphyria.

Inherited, loss-of-function mutations in any of the heme biosynthesis enzymes downstream of ALAS (or gain-of-function mutations in ALAS) cause a group of diseases collectively referred to as porphyrias. These conditions are marked by characteristic clinical features – neurovisceral symptoms, skin lesions, or both (*Karim et al., 2015*). While potentially attributable at least in part to heme deficiency, the primary etiology of these symptoms entails a buildup of pathway intermediates due to the bottleneck effect created by underlying mutations. ALA, and possibly also its pyrrole derivative porphobilinogen (PBG), is neurotoxic (*Pierach and Edwards, 1978*; *Adhikari et al., 2006*; *Felitsyn et al., 2008*; *Bissell et al., 2015*). Porphyrins cause cutaneous abnormalities by virtue of their photosensitizing properties. Though generated primarily in the liver or bone marrow, the major sites of heme biosynthesis, they are eventually deposited in other tissues including the dermis. There, they readily absorb light to enter an excited triplet state that reacts with oxygen to produce singlet oxygen. This in turn results in oxidative damage, such as lipid peroxidation and DNA damage, that can ultimately lead to cell death (*Poh-Fitzpatrick, 1986*). Thus, sunlight, or even bright indoor light, quickly damages exposed skin in many porphyria patients. Clinically, porphyrin-mediated phototoxicity manifests as edema, blistering skin lesions, and in extreme cases, disfiguring scarring and/or tissue loss (*Balwani and Desnick, 2012*; *Karim et al., 2015*).

A subset of porphyrias classified as 'acute' present with sudden and potentially life-threatening attacks characterized by severe abdominal pain and neurological symptoms ranging from anxiety and confusion to seizures or paralysis (*Balwani and Desnick, 2012*; *Karim et al., 2015*). These episodes, which can last for weeks, are triggered by drugs (e.g., barbiturates), hormonal changes, dieting/fasting, and other factors that induce hepatic ALAS expression or activity. Accordingly, they are treated with therapies that effect ALAS downregulation. Carbohydrate loading reduces ALAS levels via insulin signaling and an antagonistic effect on its transcriptional coactivator PGC-1α (peroxisome

proliferator-activated receptor γ coactivator 1α) (*Scassa et al., 2004*; *Handschin et al., 2005*). This is sometimes effective in ameliorating mild attacks. More severe cases are treated with intravenous heme, which downregulates ALAS expression through a feedback inhibition mechanism (*Bonkowsky et al., 1971*; *Ponka, 1997*). A small interfering RNA therapy targeting ALAS decreased plasma ALA and PBG levels in a mouse model of acute porphyria (*Yasuda et al., 2014*), and entered clinical trials in 2015 (*Alnylam Pharmaceuticals, 2016*, NCT02452372).

Given their toxic effects in porphyrias, it is interesting to note that porphyrins accumulate under physiological circumstances in some organisms. Porphyrins or their derivatives are pigments in the wing feathers of owls (*With, 1978*; *Weidensaul, 2011*), the brilliant crimson flight feathers of turacos (*With, 1957*), and numerous invertebrate lineages including earthworms, molluscs, and deep-sea medusae (*Kennedy, 1975*). It has been known for close to a century that high levels of porphyrins are present in the rodent Harderian gland (*Derrien and Turchini, 1924*), as well as in multiple tissues of the fox squirrel *Sciurus niger* (*Turner, 1937*), though the significance of these observations remains mysterious. Porphyrins are also produced by bacteria such as *Propionibacterium acnes*, a commensal skin microbe that contributes to the pathogenesis of acne (*Lee et al., 1978*). In summary, porphyrins appear to have important, but often uncharacterized functions independent of their more well-known roles as precursors to their metal-coordinated counterparts (e.g., heme and the magnesium porphyrin chlorophyll).

Here we extend previous biochemical studies documenting physiological porphyrin biosynthesis in the planarian *Girardia* (formerly *Dugesia*) *dorotocephala* (*MacRae, 1956*; *1959*; *1961*; *1963*) by showing the related planarian species *Schmidtea mediterranea* uses the first 3 enzymes in the heme biosynthesis pathway to generate porphyrins in its subepithelial pigment cells. We further demonstrate that porphyrins sensitize *S. mediterranea* to intense visible light, causing pigment cell loss in animals subjected to prolonged light exposure. This response is accelerated with starvation, echoing the connection between dieting or fasting and the symptomatic attacks experienced by acute porphyria patients. In addition to serving as a popular model organism for regeneration research (*Gentile et al., 2011*; *Reddien, 2013*; *Adler and Sánchez Alvarado, 2015*), our observations establish planarians as an invertebrate model for in vivo studies of porphyrin photosensitization, as well as metabolic inputs into the pathophysiology of acute porphyrias.

## Results

### Prolonged light exposure induces bodily depigmentation in *S. mediterranea*

This line of research originated with a serendipitous discovery in an undergraduate, general education course – sunlight exposure causes depigmentation of both regenerating and intact planarians (*Figure 1—figure supplement 1*). Infrared (IR) and ultraviolet B (UVB) radiation were neither necessary nor sufficient to induce depigmentation under conditions we tested in follow-up experiments (*Figure 1—figure supplement 2*). In contrast, we were able to reproduce this response with intense visible light (*Figure 1A,B* and *Figure 1—figure supplement 3*). Just over half of light-exposed animals (51%; n = 864 analyzed in 18 independent experiments) developed one or more small tissue lesions on their dorsal surface; 4% lysed. Apart from these defects and their lack of bodily pigmentation, depigmented animals were indistinguishable from controls, exhibiting normal movement (*Video 1*), touch-responsiveness (*Video 2*), feeding behavior (*Figure 1—figure supplement 4A*), and regenerative ability (*Figure 1—figure supplement 4B*). Depigmented animals repigmented when light exposure stopped (*Figure 1—figure supplement 4C–E*).

Two observations indicated light-induced depigmentation was not a simple photobleaching effect. First, depigmentation was non-uniform (*Figure 1C–E*), whereas direct photobleaching should result in relatively even pigment loss. Second, animals subjected to a 24-hr pulse of light exposure continued to depigment for several days after light exposure stopped (*Figure 1—figure supplement 5*). We conclude that prolonged exposure to intense visible light is sufficient to induce a reversible depigmentation response in *S. mediterranea*.

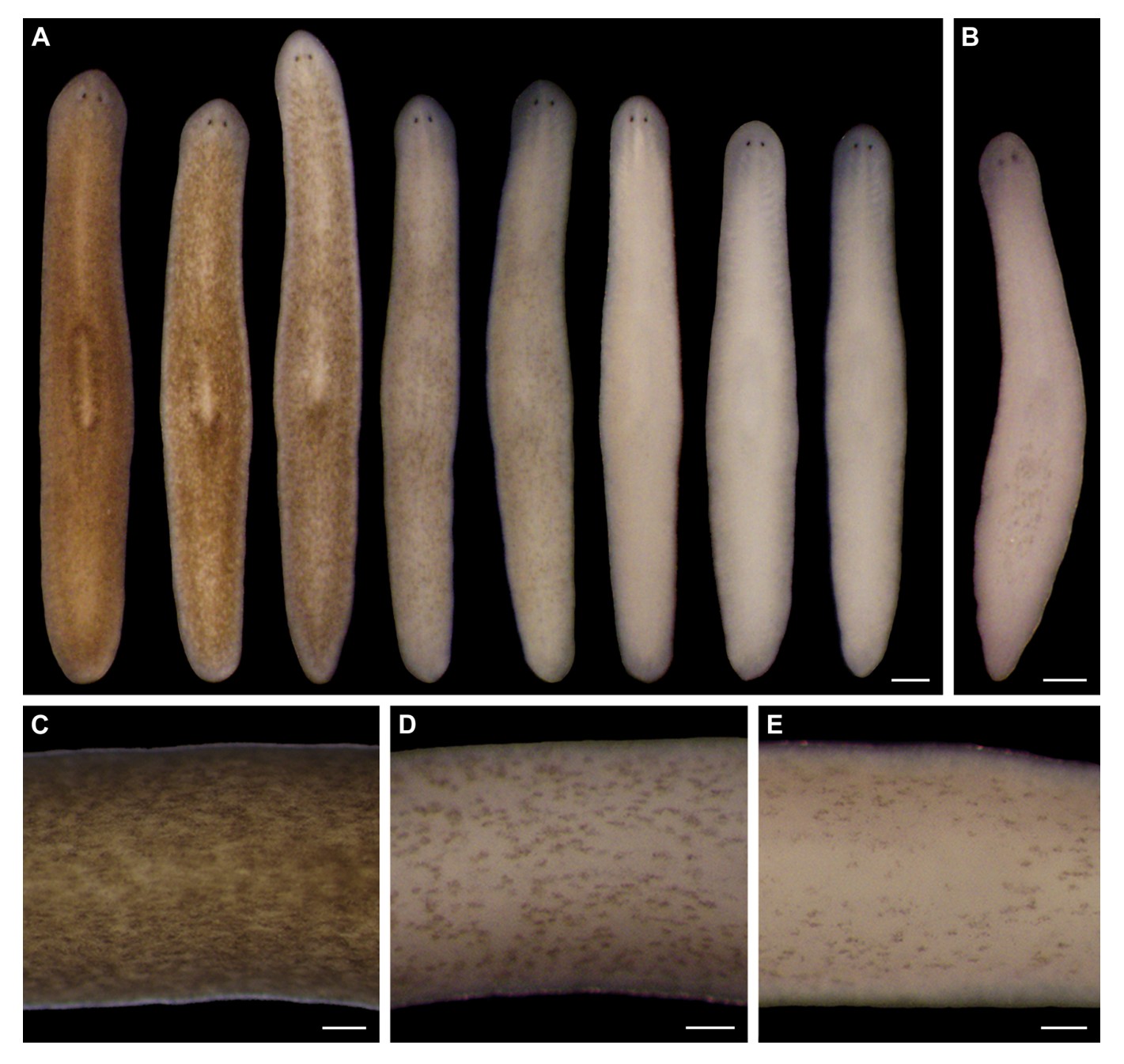

**Figure 1.** Light-induced depigmentation in *S. mediterranea*. (**A**) Animals exposed to visible light (incident intensity = 5000 lux; see figure supplement 3 for spectrum) exhibit progressive loss of bodily pigmentation. Images show a single live animal photographed (left to right) at time 0 and immediately following each of a series of intermittent light exposure and recovery periods (final timepoint = 10 days; see Materials and methods for details). Continuous exposure results in 100% lethality at this intensity. (**B**) Ventral surface of a representative depigmented animal. (**C–E**) Close-ups of control (**C**) and light-exposed (**D,E**) animals demonstrating non-uniform pigment loss (photographs in **D** and **E** correspond to 4[th] and 5[th] images from left in **A**). Scale bars: **A,B** = 500 µm; **C–E** = 200 µm.

The following figure supplements are available for figure 1:

**Figure supplement 1.** Sunlight-induced depigmentation.

**Figure supplement 2.** IR and UVB radiation are neither necessary nor sufficient to induce depigmentation.

*Figure 1 continued*

**Figure supplement 3.** Incident spectrum for visible light exposure.
**Figure supplement 4.** Feeding, regeneration, and repigmentation in depigmented animals.
**Figure supplement 5.** Depigmentation is not due to direct photobleaching.

### *S. mediterranea* produces an ommochrome body pigment

To gain insight into the mechanisms of light-induced depigmentation, we sought to identify the *S. mediterranea* body pigment. The planarian *Dugesia ryukyuensis*, which also exhibits a brown color, produces a body pigment previously classified as an ommochrome on the basis of its absorption spectrum (*Hase et al., 2006*). We found that a *S. mediterranea* body pigment co-purifying with RNA (Materials and methods), like the *D. ryukyuensis* ommochrome, exhibited characteristic, local absorption maxima near 367 and 463 nm; a mock pigment purification from depigmented animals resulted in minimal absorbance (*Figure 2A*).

Ommochromes are tryptophan-derived pigments produced in the eyes of many insect species, including *Drosophila*. Their biosynthesis involves an evolutionarily conserved pathway consisting of 4 steps (*Ryall and Howells, 1974*). We searched *S. mediterranea* genomic and EST databases (*Labbé et al., 2012*; *Robb et al., 2015*; *Zhu et al., 2015*) for candidate ommochrome biosynthesis enzymes using a reciprocal BLAST approach (Materials and methods) and identified a total of 7 genes corresponding to the first 3 steps in the pathway (*Figure 2B*; *Figure 2—source data 1*. Apparent absence of the terminal enzyme, phenoxazinone synthetase, is consistent with prior analyses of other animal genomes, including those of ommochrome-producing species (e.g., *Croucher et al., 2013*), and may reflect non-enzymatic oxidation of 3-hydroxykynurenine (*Wiley and Forrest, 1981*).

We used whole-mount in situ hybridization (WISH) to analyze where these genes are expressed (*Figure 2C*). *Smed-kynurenine 3-monooxygenase-1 (KMO-1)*, 1 of 3 *S. mediterranea KMO* homologs, exhibited pigment cell-specific or enriched expression – staining was evident in cells distributed across the entire surface of the animal, but excluded from the unpigmented regions of the photoreceptors. Nearly all *KMO-1*-expressing cells were lost during light-induced depigmentation (see below). To our knowledge, this represents the first marker for a planarian pigment cell outside of the melanin-producing cells of the eye cups (*Lapan and Reddien, 2011*). The second *KMO* homolog, *Smed-kynurenine 3-monooxygenase-2 (KMO-2)*, also appeared to be expressed in body pigment cells (in addition to the gut), while the third, *KMO-3*, was expressed in a small group of cells of unknown function clustered anterior to the pharynx. We also characterized *S. mediterranea* homologs of 2 different enzymes capable of oxidizing tryptophan to N-formylkynurenine, the first reaction

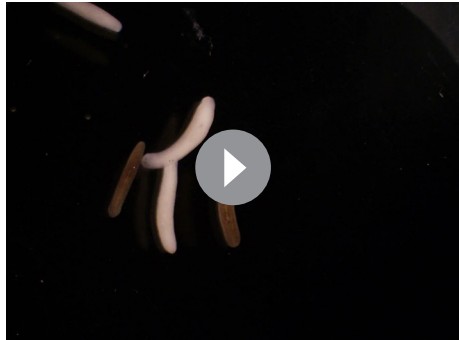

**Video 1.** Depigmented animals exhibit normal movement. Control animals maintained under standard laboratory conditions were filmed next to depigmented animals shortly after the conclusion of light exposure.

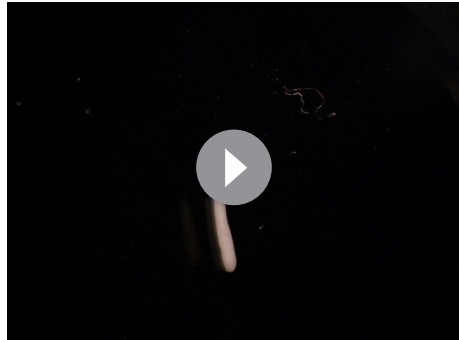

**Video 2.** Depigmented animals exhibit normal touch responsiveness. Like controls, depigmented animals change direction in response to the touch of a pipet tip.

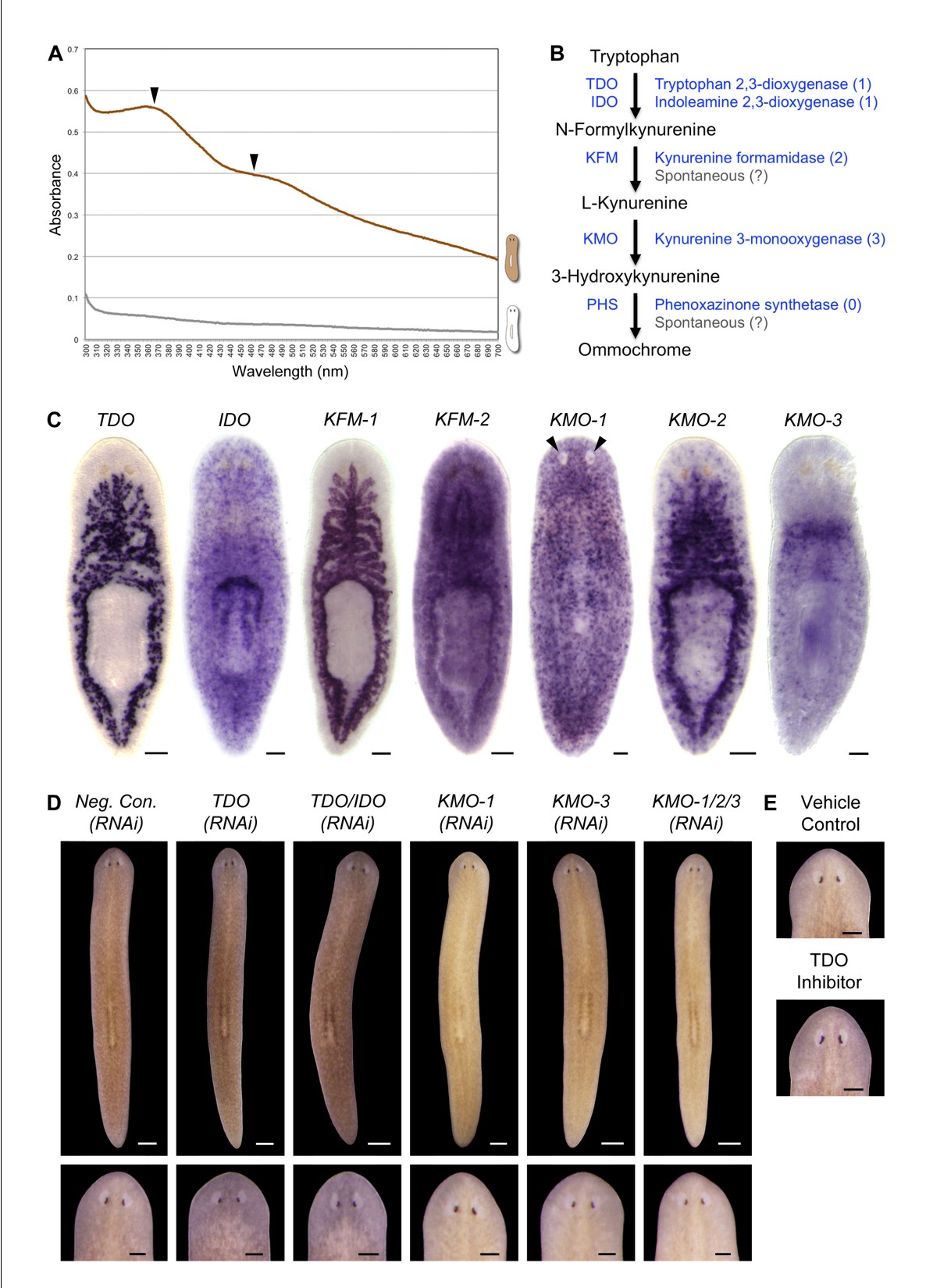

**Figure 2.** *S. mediterranea* produces an ommochrome body pigment. (**A**) Absorbance spectra of body pigment purified from control animals (brown line) or mock purified from depigmented animals (grey line). Arrowheads denote local maxima at 367 and 463 nm, characteristic of ommochrome
*Figure 2 continued on next page*

*Figure 2 continued*

pigments. (B) Ommochrome biosynthesis pathway. Numbers in parentheses to the right of each enzyme denote the number of *S. mediterranea* homologs identified via reciprocal BLAST (Materials and methods; source data 1). Enzyme abbreviations are shown to the left. (C) Whole-mount in situ hybridizations for candidate ommochrome biosynthesis genes. Note absence of *KMO-1*-expressing cells from unpigmented regions of the photoreceptors (arrowheads) and lower numbers of these cells over the pharynx (center), an area of reduced pigmentation. (D) RNAi phenotypes for candidate ommochrome biosynthesis genes (results not shown for genes that did not generate a visible phenotype). Intact animals (top) were administered a total of 12 RNAi feedings over 3.5 weeks and photographed 3 days after the final feeding. Regenerated animals (bottom) were amputated at this timepoint to remove cephalic and caudal tissue. The resulting trunk fragments were allowed to regenerate for 2 weeks, administered 3 further RNAi feedings, and photographed at 21 days post-amputation. (E) Animals were placed in solutions containing the tryptophan 2,3-dioxygenase inhibitor 680C91 (0.7 µM final concentration) or a vehicle (ethanol) control immediately after cephalic amputation, and photographed after 16 days of regeneration. Scale bars: C = 100 µm; D = 500 µm (top), 200 µm (bottom); E = 200 µm.

The following source data is available for figure 2:

**Source data 1.** *S. mediterranea* ommochrome biosynthesis enzymes.

in ommochrome biosynthesis. *Smed-tryptophan 2,3-dioxygenase (TDO)* was expressed in the gut. *Smed-indoleamine 2,3-dioxygenase (IDO)* exhibited an expression pattern resembling that of *KMO-1*, but we have not determined whether these transcripts are present in the same cell type. The 2 remaining pathway genes, *Smed-kynurenine formamidase-1* and *-2 (KFM-1* and *-2)*, showed specific or enriched expression in the central nervous system and/or gut.

We next used RNA interference (RNAi) to assess the functions of these genes. In multiple cases, this resulted in a noticeable change in body color (*Figure 2D*). *KMO-1(RNAi)* animals developed a yellow hue. This phenotype was also evident in *KMO-3(RNAi)* animals; simultaneous knockdown of all 3 *KMO* homologs did not alter the color change. RNAi knockdown of *TDO* resulted in a charcoal grey color that was particularly apparent in the regeneration blastema, the mass of stem cell-derived tissue that forms at sites of amputation (*Figure 2D*, bottom). Treatment with a pharmacological TDO inhibitor phenocopied the *TDO(RNAi)* effect (*Figure 2E*). Although IDO catalyzes the same reaction as TDO, we detected at most a subtle color change in *IDO(RNAi)* animals, and did not observe an enhanced phenotype following double knockdown. No RNAi phenotypes were apparent for the 2 *KFM* homologs when targeted individually or in combination. These results are consistent with the prior identification of *TDO (vermilion)* and *KMO (cinnabar)*, but not *KFM* eye color mutants in *Drosophila* (*Searles and Voelker, 1986*; *Warren et al., 1996*), as well as the potential for non-enzymatic generation of kynurenine from formylkynurenine (*Linzen, 1974*).

It is not clear why inhibiting genes in this pathway resulted in color changes rather than depigmentation. This could be due to incomplete gene knockdown, the presence of another body pigment in analogy with the *Drosophila* eye, or accumulation of colored intermediates in ommochrome biosynthesis (for instance, kynurenine imparts a yellow color to the eyes of deep-sea fish; *Thorpe et al., 1992*). Nevertheless, when taken together with the absorption spectrum of purified body pigment and the pigment cell expression of *KMO-1*, the color changes in RNAi animals strongly suggest *S. mediterranea* produces an ommochrome body pigment.

## *S. mediterranea* pigment cells produce porphyrins

Ommochromes are not known to act as photosensitizers, but porphyrins, the reported body pigment in the planarian *G. dorotocephala* (*MacRae, 1956*; *1959*; *1961*; *1963*), are. We therefore sought to determine whether porphyrins are produced in *S. mediterranea*, and if so, whether they might contribute to the depigmentation response we observed in light-exposed animals.

A classic biochemical signature of porphyrins is their bright red/pink fluorescence under black light (UVA). This trait is used to age owls on the basis of the porphyrin content of their flight feathers (*Weidensaul et al., 2011*), and to detect excreted porphyrins in the urine of porphyria patients (*Balwani and Desnick, 2012*). We reproduced previous results showing intense red fluorescence in fixed *G. dorotocephala* at 400–440 nm excitation (*MacRae, 1961*), and found the related species *Dugesia japonica* was also highly fluorescent under these conditions (*Figure 3A*). *S. mediterranea* exhibited minimal fluorescence by comparison; however, *KMO-1(RNAi)* animals showed a dramatic increase in fluorescence relative to negative controls (*Figure 3B*). To confirm this was due to porphyrins, we acid extracted whole-animal homogenates (Materials and methods) and determined

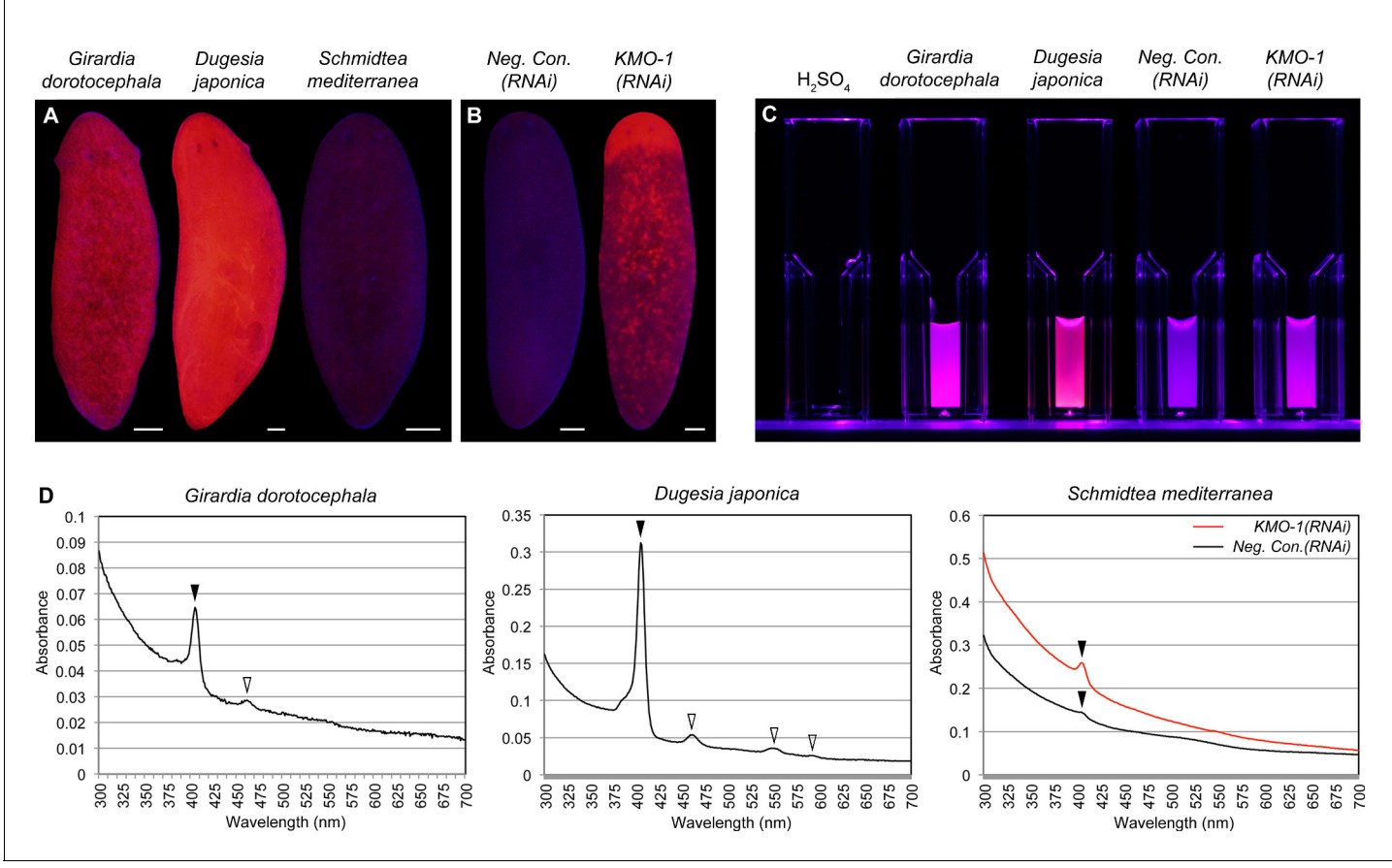

**Figure 3.** Biochemical evidence of porphyrin biosynthesis in *S. mediterranea*. (**A**) Like *G. dorotocephala*, *D. japonica* exhibits bright red fluorescence under black light (400–440 nm excitation). *S. mediterranea* exhibits negligible fluorescence by comparison. (**B**) *KMO-1(RNAi)* animals demonstrate strongly increased fluorescence relative to negative controls. The uniform fluorescence in the anterior (top) corresponds to recently regenerated tissue (animals were photographed 3.5 weeks after cephalic amputation). Fluorescence does not appear to be restricted to pigment cells, particularly within regenerated tissue; this may reflect porphyrin movement across cell membranes (*Viljoen et al., 1975*). (**C**) Whole-animal $H_2SO_4$ homogenates were photographed in plastic cuvettes over a black (400 nm) LED. (**D**) Representative absorption spectra of whole-animal homogenates. Black arrowheads denote the Soret peak (405 nm); a visible increase in the height of this peak was evident for 3/3 *KMO-1(RNAi)* homogenates relative to *Neg. Con.(RNAi)* homogenates. White arrowheads denote Q bands (459 nm for *G. dorotocephala*; 459, 549, and 584 nm for *D. japonica*). All scale bars = 200 μm.

their absorption spectra. Like *G. dorotocephala* and *D. japonica* extracts, *KMO-1(RNAi)* extracts demonstrated bright red fluorescence under black light (*Figure 3C*), as well as a characteristic porphyrin absorption spectrum (*Huang et al., 2000*) with a sharp peak (the 'Soret' band) around 400 nm (*Figure 3D*). Smaller peaks ('Q' bands) were evident within the visible region of the *G. dorotocephala* and *D. japonica* spectra, but could not be resolved for *KMO-1(RNAi)* extracts. Importantly, faint fluorescence and a very small, yet reproducible Soret peak were apparent for *S. mediterranea* controls. We conclude that *S. mediterranea* makes porphyrins like *G. dorotocephala*, but at substantially lower levels, or primarily in a non-fluorescent (e.g., reduced or metal-chelate) form. The effects of *KMO-1* knockdown further suggest that porphyrin biosynthesis occurs in pigment cells and may be suppressed by ommochromes (Discussion).

Porphyrins are physiological intermediates in heme biosynthesis. To address the hypothesis that they are produced in *S. mediterranea* pigment cells, we identified, cloned, and characterized genes in the heme biosynthesis pathway (*Figure 4A*; *Figure 4—source data 1*), using the same basic approach as described above for the ommochrome pathway. Strikingly, *S. mediterranea* homologs of *ALAS*, *ALA dehydratase (ALAD)*, and *PBG deaminase (PBGD)*, enzymes catalyzing the first 3 reactions in metazoan heme biosynthesis, were highly expressed in pigment cells, while the 5 remaining genes in the pathway were expressed predominantly or exclusively in other cell types (*Figure 4B*). In

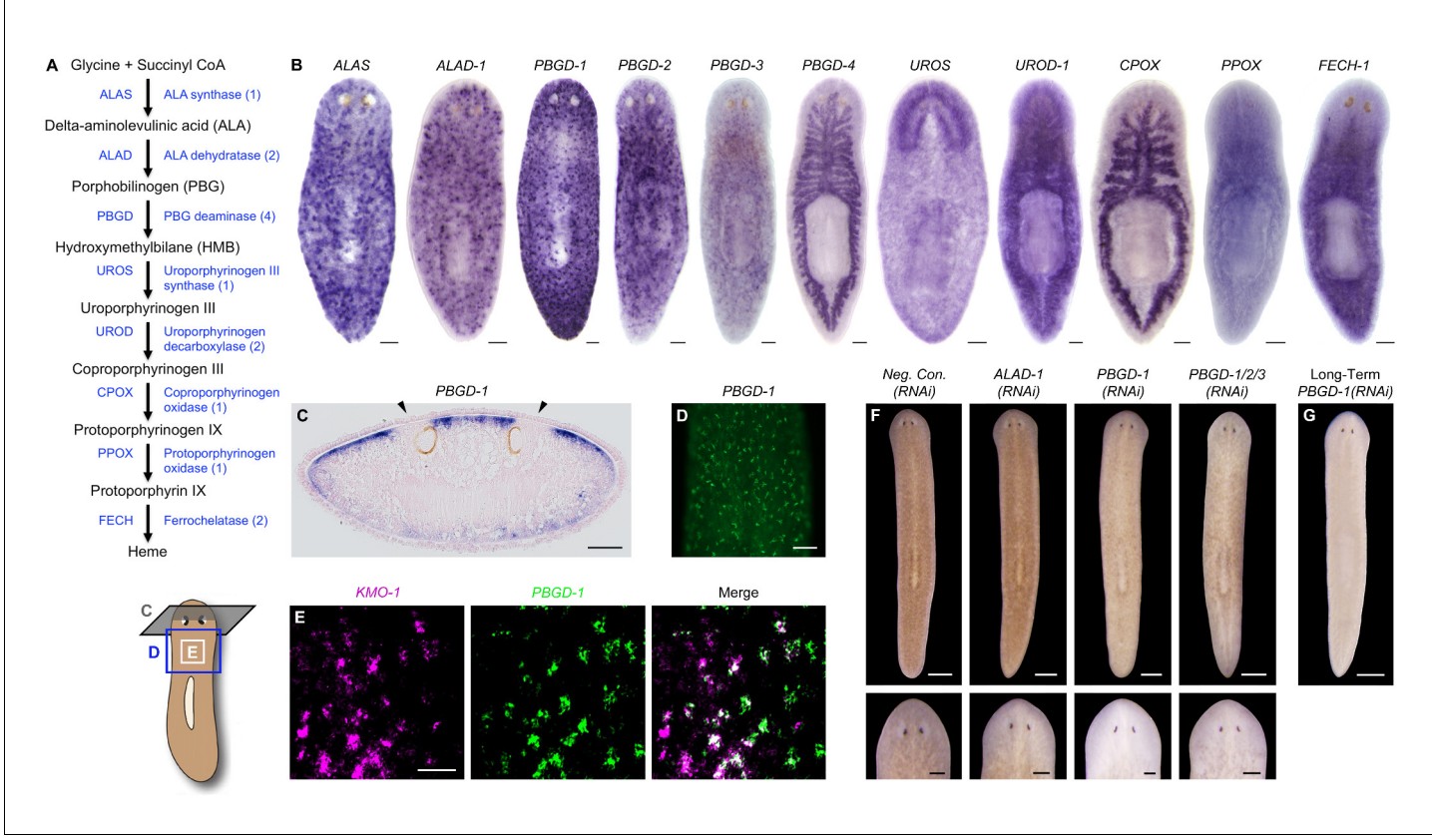

**Figure 4.** Genetic evidence of porphyrin biosynthesis in *S. mediterranea* pigment cells. (**A**) Heme biosynthesis pathway. Numbers in parentheses to the right of each enzyme denote the number of *S. mediterranea* homologs identified via reciprocal BLAST (Materials and methods; source data 1). Enzyme abbreviations are shown to the left. (**B**) Whole-mount in situ hybridizations for porphyrin/heme biosynthesis genes. *ALAS, ALAD-1*, and *PBGD-1, -2*, and *-3* expression patterns resemble that of *KMO-1* (*Figure 2C*). *PBGD-4, UROD-1, CPOX*, and *FECH-1* exhibit enriched expression in the gut. *UROS* is highly expressed in the brain. (**C**) 10 µm transverse section from the anterior of a *PBGD-1*-labeled animal, stained with nuclear fast red. *PBGD-1* expression is subepithelial, higher on the dorsal surface (matching that surface's higher level of pigmentation), and excluded from the unpigmented regions just above the photoreceptors (arrowheads). (**D**) *PBGD-1* fluorescent in situ hybridization (FISH), showing dendritic morphology of pigment cells. (**E**) Double FISH showing overlap in *KMO-1* and *PBGD-1* expression. Over 90% of *KMO-1*-positive cells were co-labeled with *PBGD-1* and vice versa (n = 11 animals analyzed by confocal microscopy). (**F**) RNAi phenotypes for porphyrin/heme biosynthesis genes (results not shown for genes that did not generate a visible phenotype, or that generated phenotypes unrelated to pigmentation – see figure supplement 1). Intact animals (top) were administered a total of 12 RNAi feedings over 3.5 weeks and photographed 3 days after the final feeding. Regenerated animals (bottom) were amputated at this timepoint to remove cephalic and caudal tissue. The resulting trunk fragments were allowed to regenerate for 2 weeks, administered 3 further RNAi feedings, and photographed at 21 days post-amputation. (**G**) Long-term RNAi feeding for *PBGD-1* leads to complete loss of bodily pigmentation. This animal was from a group fed a total of 50 times over 6 months, with periodic amputation to increase numbers. Scale bars: **B–D** = 100 µm; **E** = 50 µm; **F** = 500 µm (top), 200 µm (bottom); **G** = 500 µm.

The following source data and figure supplement are available for figure 4:

**Source data 1.** *S. mediterranea* porphyrin/heme biosynthesis enzymes.

**Figure supplement 1.** Additional RNAi phenotypes for porphyrin/heme biosynthesis genes.

agreement with histological analyses of *Dugesia gonocephala* chromatophores (*Palladini et al., 1979*), *PBGD-1*-expressing cells were located just beneath the epithelium (*Figure 4C*) and exhibited a dendritic morphology also characteristic of other pigment-producing cell types (*Figure 4D*). Double fluorescent in situ hybridization (dFISH) confirmed *Smed-PBGD-1 (PBGD-1)* was co-expressed with *KMO-1* (*Figure 4E*).

*ALAS, ALAD*, and *PBGD*, but not downstream enzymes, exhibited RNAi phenotypes indicative of a role in pigment biosynthesis. Specifically, while *ALAS(RNAi)* animals developed morphological

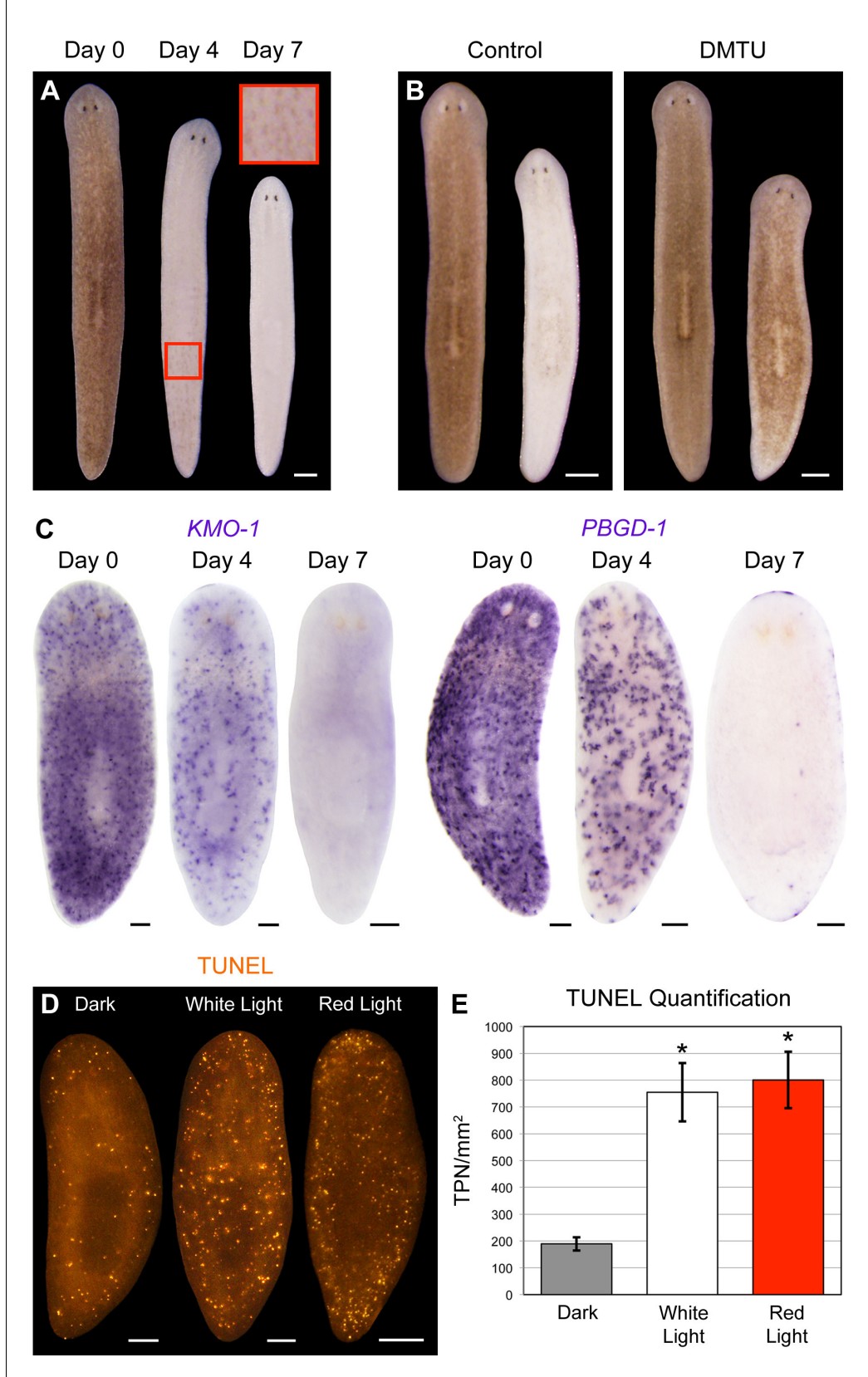

**Figure 5.** Visible light exposure causes pigment cell loss. (**A**) Red light (625 nm LED) is sufficient to induce full bodily depigmentation. Inset shows a magnified view of the dorsal surface, brightness, contrast, and gamma-enhanced to highlight remaining pigment cells. No animals developed lesions or

*Figure 5 continued on next page*

*Figure 5 continued*

lysed under these conditions (n = 345 analyzed in 23 independent experiments). (**B**) DMTU inhibits light-induced depigmentation. Representative control and DMTU-treated animals were photographed before and after white light exposure (left and right in each panel, respectively). DMTU treatment (10 mM final concentration) was initiated 5 days prior to the start of light exposure and continued for the duration of the experiment. Total light exposure time = 72 hr (24- and 48-hr exposures were separated by a 24-hr dark recovery). (**C**) Light-induced depigmentation is due to pigment cell loss. Images show representative red light-exposed animals fixed at the indicated times and labeled with *KMO-1* or *PBGD-1* riboprobes. (**D**) Light-induced cell death visualized by whole-mount TUNEL. Light-exposed animals were fixed 12 hr after a 24-hr exposure. (**E**) Quantitative analysis of TUNEL results. The number of TUNEL-positive nuclei (TPN)/mm$^2$ was averaged over 3 independent experiments (n = total of 38 dark, 31 white and red light-exposed animals). Error bars = +/- s.e.m. *p-value <1 x 10$^{-4}$ for two-tailed student's t-test comparing light-exposed animals with controls. Scale bars: **A** = 300 µm; **B** = 500 µm; **C,D** = 100 µm.

The following figure supplement is available for figure 5:

**Figure supplement 1.** Antioxidants and hypoxia inhibit light-induced depigmentation.

defects culminating in 100% lethality, they became lighter than negative controls prior to dying (*Figure 4—figure supplement 1A*). *PBGD-1(RNAi)* animals were viable and showed a more pronounced reduction in pigmentation (*Figure 4F*, top), eventually turning completely white with sustained RNAi feeding (*Figure 4G*). Biochemical analysis confirmed ommochrome loss (*Figure 4—figure supplement 1B*). Following amputation, *PBGD-1(RNAi)* animals failed to produce new body pigment in the blastema (*Figure 4F*, bottom). *ALAD-1* knockdown, while not effecting a pigmentation change in uncut animals, also disrupted pigment biosynthesis in newly regenerated tissue. No changes in pigmentation were evident in either intact or regenerating animals following knockdown of the remaining heme biosynthesis enzymes, though RNAi phenotypes unrelated to pigmentation and 100% lethality were observed for *PBGD-4* and *Smed-uroporphyrinogen decarboxylase-1 (UROD-1)* (*Figure 4—figure supplement 1A*). These defects, as well as the lethality associated with *ALAS* knockdown, could in theory be due to heme deficiency. However, many organisms readily absorb heme from their diets (*Rao et al., 2005*; *Shayeghi et al., 2005*), making alternative explanations possible (e.g., noncanonical functions for pathway enzymes; *Greenbaum et al., 2003*; *Zhang et al., 2014*).

To summarize, we found the first 3 enzymes in heme biosynthesis are expressed in the body pigment cells of *S. mediterranea* and are required for pigmentation, while UROS and the remaining enzymes in the pathway are not expressed (or are limiting) in this cell type, and are dispensable for pigment biosynthesis. These results imply that *S. mediterranea* pigment cells share a metabolic bottleneck with the erythroid cells of individuals suffering from the rare, autosomal recessive disorder congenital erythropoietic porphyria (CEP) (*Xu et al., 1996*). Patients with this condition, also known as Gunther's disease, have reduction-of-function mutations in uroporphyrinogen III synthase (UROS), the 4$^{th}$ enzyme in heme biosynthesis, and consequently produce high levels of porphyrins in their erythroid cells (Discussion).

## Porphyrins photosensitize *S. mediterranea* pigment cells

Some of the porphyrin molecules generated by CEP patients accumulate in the skin, causing severe cutaneous photosensitivity usually beginning in infancy (*Xu et al., 1996*; *Balwani and Desnick, 2012*; *Karim et al., 2015*). Porphyrin-mediated photosensitization is observed to varying degrees in other porphyrias as well and has been clinically exploited in photodynamic therapy (PDT). This technique entails administration of one or more photosensitizing compounds followed by irradiation with a wavelength of light absorbed by the sensitizer(s). Red light is used in porphyrin-based PDT because it is transmitted by epithelial tissues but absorbed by porphyrins. The end result is production of singlet oxygen in targeted (e.g., tumor) cells, leading to cell death (*Agostinis et al., 2011*).

Consistent with a porphyrin-based mechanism, red light (625 nm) was sufficient to induce full bodily depigmentation (*Figure 5A*), while the antioxidants dimethylthiourea (DMTU) and ascorbic acid exerted inhibitory effects (*Figure 5B* and *Figure 5—figure supplement 1A*). Hypoxia was also protective (*Figure 5—figure supplement 1B*). We further determined that depigmentation was due to pigment cell loss – light exposure eliminated *KMO-1/PBGD-1*-positive cells, with no apparent decrease in expression of these genes within remaining pigment cells at intermediate timepoints (*Figure 5C*). Both white light and red light induced a significant increase in cell death, as measured

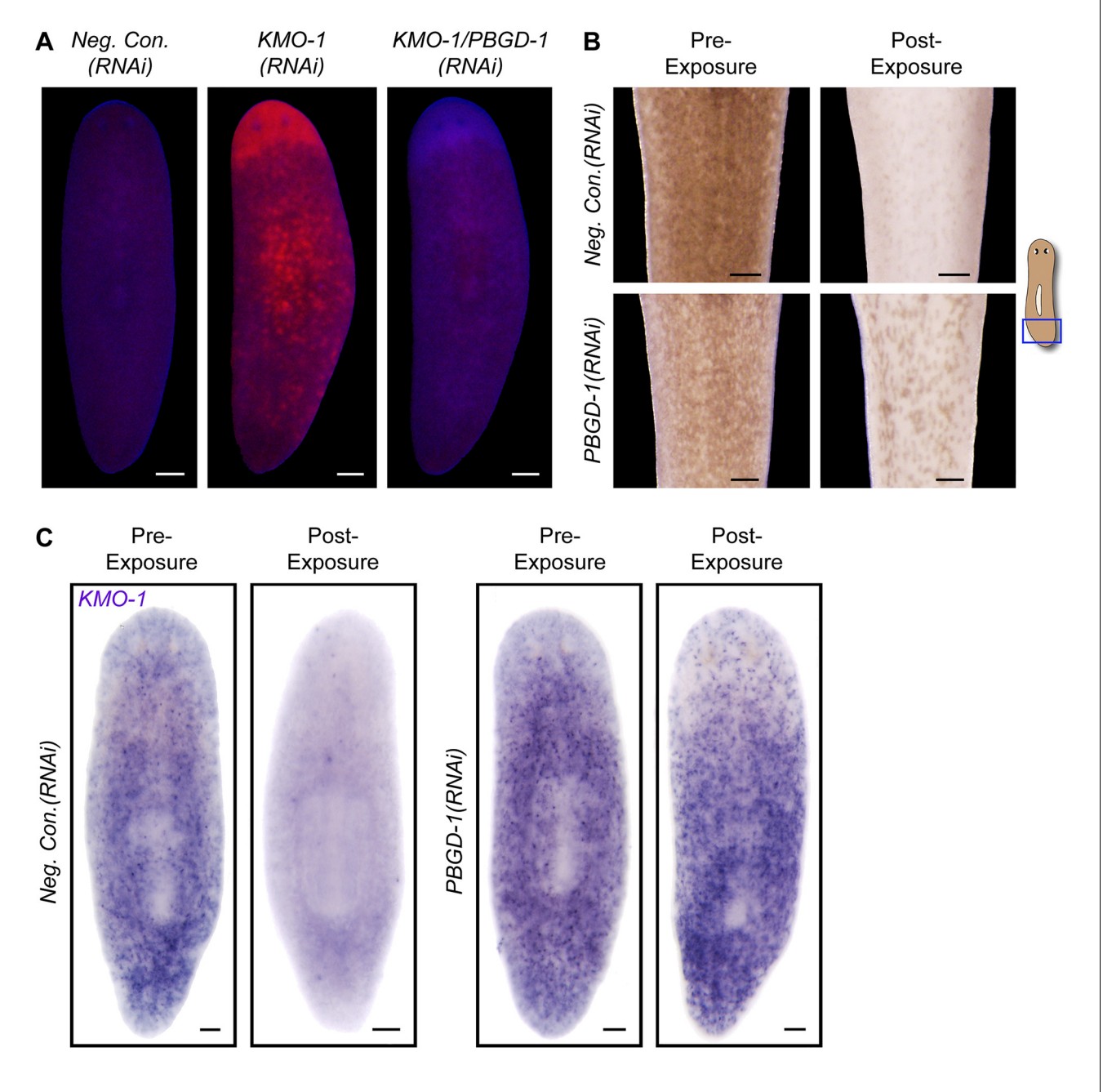

**Figure 6.** Porphyrins mediate light-induced pigment cell loss. (**A**) *PBGD-1* knockdown suppresses the porphyrin fluorescence observed in *KMO-1(RNAi)* animals. The difference in appearance of anterior tissues (top) between negative controls and *KMO-1/PBGD-1(RNAi)* animals is a consequence of the latters' failure to repigment newly regenerated tissue (animals were amputated 3.5 weeks prior to photographing). See figure supplement 1 for additional controls. (**B**) *PBGD-1* knockdown suppresses light-induced depigmentation. Animals were photographed before and after 48 hr of red light exposure. Note the greater pigmentation in *PBGD-1(RNAi)* animals after exposure, despite their lower initial pigmentation. For reasons that are presently unclear, this effect was restricted to the posterior, which typically depigments at a lower rate than anterior tissues. (**C**) *PBGD-1* knockdown suppresses light-induced pigment cell loss. A *KMO-1* riboprobe was used to visualize pigment cells in animals fixed before and after 7 days of continuous red light exposure. Scale bars: **A**,**B** = 200 μm; **C**= 100 μm.

The following figure supplement is available for figure 6:

**Figure supplement 1.** Porphyrin fluorescence controls.

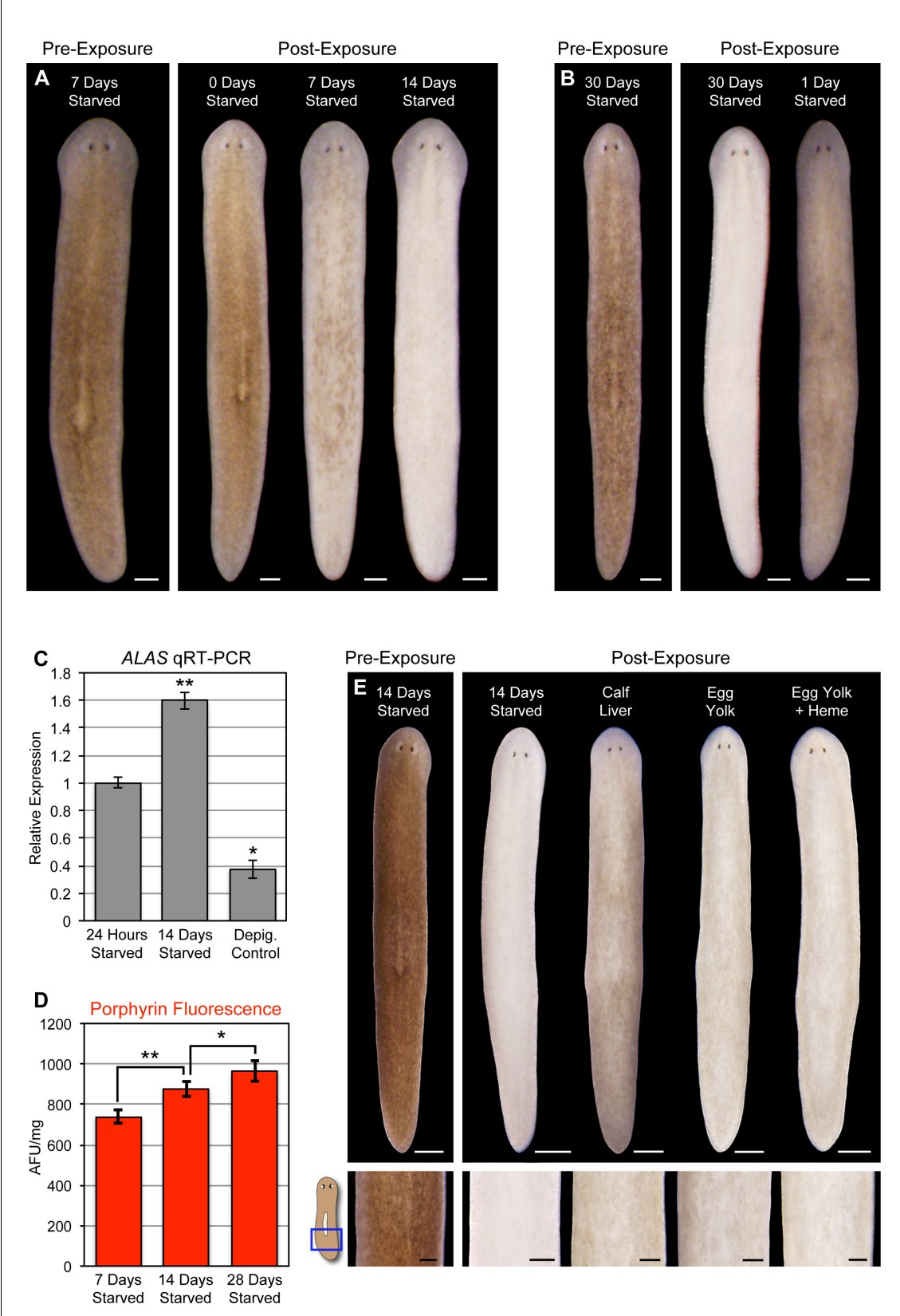

**Figure 7.** Starvation induces porphyrin biosynthesis and acute photosensitivity. (**A**) Animals were fed 4 times in 1 week with dyed calf liver and then starved as indicated prior to 72 hr of red light exposure. Representative animals were photographed pre-exposure and 72 hr after the conclusion of
*Figure 7 continued on next page*

*Figure 7 continued*

light exposure. (B) Animals given a single feeding after 29 days of starvation and light exposed 24 hr later showed far less depigmentation than 30 day-starved animals. Post-exposure photographs were taken 24 hr after the conclusion of light exposure, as full depigmentation was already apparent in 30 day-starved animals. (C) qRT-PCR analysis of *ALAS* expression. The fold change relative to 24 hr-starved animals was averaged over 3 biological replicates. Depigmented animals showed reduced expression, as predicted based on the *ALAS* expression pattern (*Figure 4B*). Error bars = +/- standard deviation. **p-value <0.001 for two-tailed student's t-test in comparison with 24 hr starved; *p-value <0.01. (D) Quantitative analysis of porphyrin fluorescence in *D. japonica* lysates, averaged over 10 biological replicates. AFU/mg = arbitrary fluorescence units/mg wet tissue weight. Error bars = +/- standard deviation. **p-value <1 x 10–7 for two-tailed student's t-test; *p-value <0.001. (E) Photoprotective effect of different food sources. Well-fed animals were fasted, light exposed, and photographed as in (A), with a subset re-fed as indicated after 13 days. Light exposure was initiated for all groups at 24 hr post-feeding (day 14). Scale bars: **A**,**B** = 300 µm; **E** = 500 µm (top), 200 µm (bottom).
The following figure supplement is available for figure 7:

**Figure supplement 1.** Starvation does not sensitize animals to induction of cell death by gamma radiation.

by whole-mount TUNEL (*Figure 5D,E*). Although we were unable to determine the cellular specificity of this response with existing methods, systemic induction of cell death by RNAi knockdown of a *BCL-2* homolog results in large tissue lesions and lysis (*Pellettieri et al., 2010*). These phenotypes were rarely observed following white light exposure and never observed with red light, suggesting cell death is not systemically induced in these contexts.

To directly address the hypothesis that porphyrins mediate light-induced depigmentation, we used RNAi to experimentally manipulate porphyrin levels in vivo. As noted above, *KMO-1(RNAi)* animals exhibited a strong increase in porphyrin fluorescence relative to negative controls. Simultaneous *PBGD-1* knockdown abrogated this effect (*Figure 6A*); it is unlikely this was due to decreased efficacy of *KMO-1* knockdown, because a double RNAi control with *Smed-ferrochelatase-1 (FECH-1)* did not alter fluorescence (*Figure 6—figure supplement 1*). As predicted, *KMO-1(RNAi)* animals showed increased photosensitivity, with 100% failing to survive 48 hr of light exposure, less than 1/3 the exposure time used to achieve depigmentation in controls. Simultaneous *PBGD-1* (but not *FECH-1*) knockdown fully rescued viability (n = 20 animals per condition analyzed in 2 independent experiments). We conclude that *PBGD-1* is required for porphyrin biosynthesis in pigment cells.

Because *PBGD-1(RNAi)* animals lose all bodily pigmentation in the absence of light exposure (*Figure 4G*), we were unable to make a direct comparison of light-induced depigmentation rates between fully affected animals and controls. However, we did observe partial protection from light-induced depigmentation when *PBGD-1(RNAi)* animals were light exposed prior to developing full phenotypic expressivity (*Figure 6B*). Furthermore, by using a *KMO-1* riboprobe to label pigment cells by in situ hybridization, we showed *PBGD-1* knockdown conferred protection from light-induced pigment cell loss (*Figure 6C*). We conclude that light-induced depigmentation in *S. mediterranea* is due to the photosensitizing action of porphyrins in its subepithelial pigment cells.

## Starvation triggers acute photosensitivity in *S. mediterranea*

Planarians can survive for months without feeding, undergoing an up to ~20-fold reduction in size through a tissue remodeling process involving increased cell death and decreased production of stem cell division progeny (*Pellettieri et al., 2010*; *González-Estévez et al., 2012*). During the course of this research, we noticed a correlation between how long animals were fasted prior to light exposure and the extent of photosensitivity. To document this relationship, we fed animals 4 times in 7 days, using dyed calf liver to verify that they had eaten during each feeding (as in *Figure 1—figure supplement 4A*). These relatively well-fed animals were then starved for 1, 7, 14, or 30 days before being light exposed. Depigmentation was strongly accelerated with starvation (*Figure 7A*), and reversed by a single feeding 24 hr prior to initiation of light exposure (*Figure 7B*). We considered the possibility that starvation might sensitize planarians to any inducer of cell death as a trivial explanation for these results. However, no difference in TUNEL staining was evident between 7 and 14 day-starved animals exposed to a sublethal dose of gamma irradiation (*Figure 7—figure supplement 1*), arguing against a nonspecific effect.

Reduced nutrient intake induces hepatic *ALAS* expression in mammals through the cAMP/CREB/PGC-1α pathway (*Handschin et al., 2005*). When this occurs in the presence of a heme biosynthesis

bottleneck, as in acute porphyria patients, it can result in porphyrin accumulation. We found that *ALAS* expression was likewise elevated by starvation in *S. mediterranea* (*Figure 7C*), and took advantage of the strong porphyrin fluorescence in D. *japonica* (*Figure 3A,C*) to show that fasting also leads to increased porphyrin levels in tissue homogenates (*Figure 7D*). Because calf liver, the food source used in these experiments, is rich in heme, and heme exerts feedback inhibition of *ALAS* expression (*Bonkowsky et al., 1971*; *Ponka, 1997*), we sought to determine whether this might account for the photoprotective effects of feeding. Egg yolk (another laboratory food source) was slightly less effective than liver in reversing starvation-induced photosensitivity (*Figure 7E*), consistent with this possibility. However, addition of exogenous heme had no impact. While our results do not exclude a role for dietary heme in regulation of porphyrin biosynthesis in planarians, they do suggest porphyrin levels are at least partly influenced by metabolic regulation of *ALAS* expression, as in acute porphyrias.

## Discussion

Our results indicate that *S. mediterranea* produces both ommochromes and porphyrins in its subepithelial pigment cells, and that prolonged light exposure eliminates these cells through a mechanism involving porphyrin-dependent photosensitization (*Figure 8*). As discussed below, this provides new insight into pigment biosynthesis in the Platyhelminthes (flatworms) and establishes planarians as a useful animal model for porphyria research.

### Ommochrome and porphyrin biosynthesis in flatworms

The striking variety of pigment colors and patterns in the Platyhelminthes is one of the most conspicuous characteristics of this phylum, yet only a few pigments have been biochemically or genetically identified to date. The black color of the eye cups in species including *S. mediterranea, D. ryukyuensis*, and likely *G. dorotocephala* is due to melanins (*Ness et al., 1996*; *Hase et al., 2006*; *Lapan and Reddien, 2011*; *Lambrus et al., 2015*), while the brown body color in *D. ryukyuensis* and *G. dorotocephala* has been attributed to ommochromes and porphyrins, respectively (*MacRae, 1956*; *1959*; *1961*; *1963*; *Hase et al., 2006*). It is not clear whether the latter observations reflect interspecies variation, or whether each of these species generates multiple body pigments as we propose here for *S. mediterranea*. Hemoglobin has been identified in free-living and endosymbiotic rhabdocoels as well as trematode parasites (*Young and Harris, 1973*; *Phillips, 1978*; *Jennings and Cannon, 1987*; *Kiger et al., 1998*).

Ommochromes and porphyrins have a wide range of colors in addition to brown and black. Named for their prevalence in the ommatidia of arthropod eyes, the former can also be red or yellow. Porphyrins and their derivatives not only confer the red color of hemoglobin and the green color of chlorophyll, but can sometimes be blue or purple ('porphyrin' is derived from the ancient Greek word for purple, 'porphura'). Thus, the growing evidence for production of these pigments in brown freshwater planarians also makes them logical candidates for contributing to the bright hues seen in many terrestrial flatworms and marine polyclads (e.g., *Breugelmans et al., 2012*; *Lapraz et al., 2013*; *Noreña et al., 2014*). Our initial characterization of the underlying biosynthetic pathways (*Figures 2* and *4*) will facilitate future evo-devo studies exploring the evolutionary basis for this diversity, as well as the lack of pigmentation in some cave-dwelling species (e.g., *de Souza et al., 2015*). Additionally, the observation that light-exposed *S. mediterranea* repigment when returned to a dark environment (*Figure 1—figure supplement 4C–E*), presumably through replacement of lost pigment cells (*Figure 5C*), sets the stage for mechanistic analyses of pigment cell differentiation.

The RNAi phenotypes of *KMO-1* and *PBGD-1* further imply biochemical or genetic interactions between ommochromes and porphyrins or their underlying biosynthetic pathways. *KMO-1* knockdown resulted in increased porphyrin fluorescence (*Figure 3B,C*) and heightened photosensitivity, while *PBGD-1* knockdown caused loss of all visible bodily pigmentation (*Figure 4G*; *Figure 4—figure supplement 1B*). One possible explanation for these results is that ommochromes and porphyrins are jointly used as precursors in a downstream biosynthetic step to generate a single, brown body pigment, or that these molecules physically interact in a manner that quenches porphyrin fluorescence and is required for ommochrome maintenance. We are unaware of documented ommochrome-porphyrin interactions, but porphyrins can form complexes with melanins and this quenches

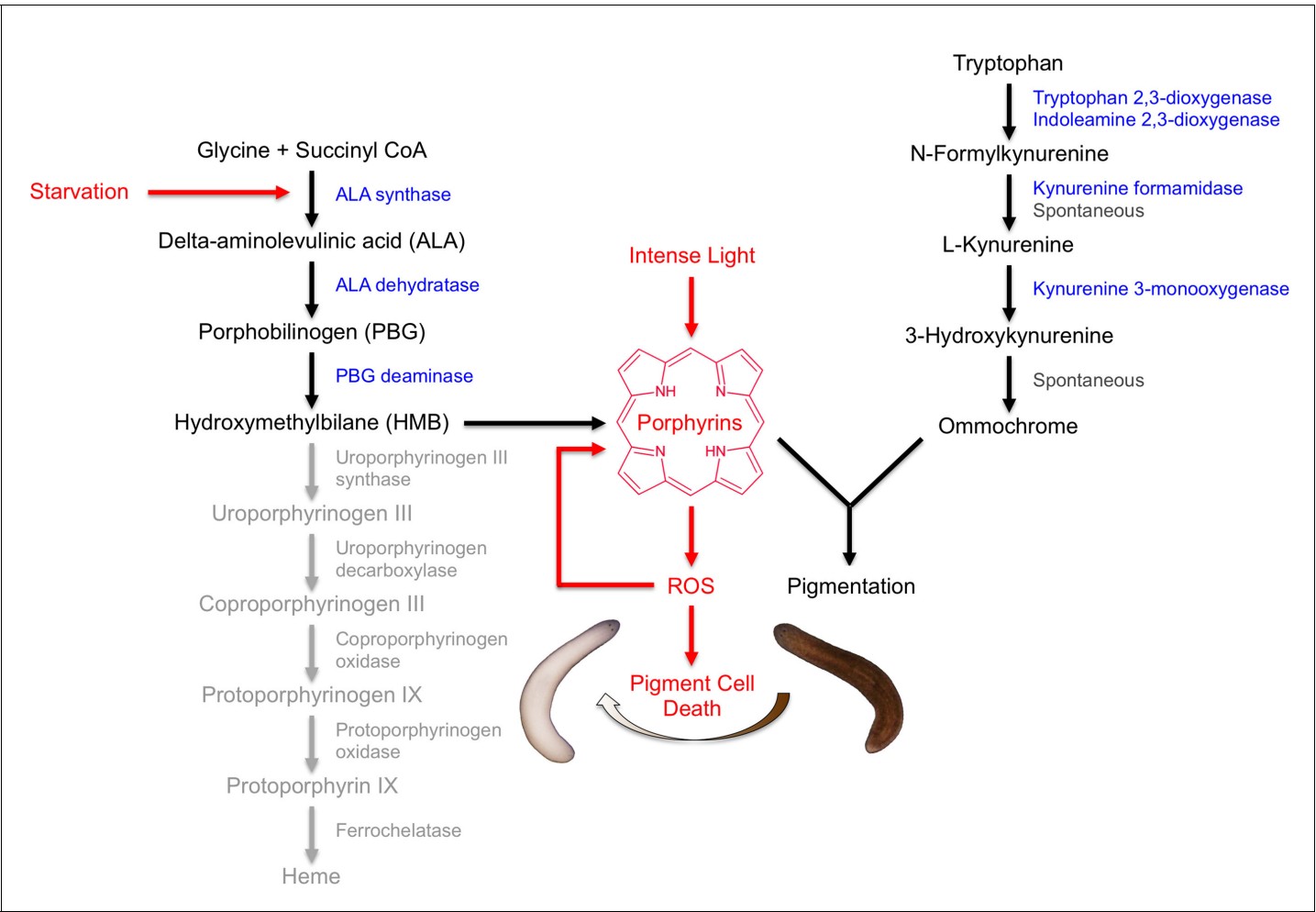

**Figure 8.** Model of *S. mediterranea* pigment biosynthesis and photosensitivity. Body pigment cells produce both porphyrins and ommochrome, using the indicated biosynthetic pathways (black arrows). Together, these molecules confer the normal brown body color, possibly as a result of a physical interaction or joint use as precursors in a downstream biosynthetic step. Porphyrin formation entails PBG deaminase-dependent HMB synthesis; limiting (or absent) expression of downstream enzymes in the heme biosynthesis pathway (grey) leads to non-enzymatic cyclization of HMB to form uroporphyrinogen I, as in the erythroid cells of CEP patients (Discussion). Uroporphyrinogen I is a reduced porphyrin that can undergo spontaneous oxidation to form the potent photosensitizer uroporphyrin I. Intense light exposure causes porphyrin molecules to generate reactive oxygen species (ROS). This has the potential to initiate a positive feedback loop in which ROS drive further uroporphyrinogen I oxidation, leading to oxidative stress and pigment cell death. Starvation accelerates this response, at least partly through induction of *ALAS* expression and a consequent increase in porphyrin levels.

their fluorescence in vitro (*Ito et al., 1992*; *Losi et al., 1993*). An alternative explanation for the effects of *KMO-1* knockdown is that ommochromes maintain porphyrins in their non-fluorescent, reduced (porphyrinogen) state. This could account for the minimal porphyrin fluorescence observed in *S. mediterranea* (*Figure 3A*). Ommochromes are known to have antioxidant activity in other species (*Ostrovsky et al., 1987*; *Insausti et al., 2013*; *Romero and Martínez, 2015*). Finally, our results are consistent with a negative feedback loop in which porphyrins induce ommochrome biosynthesis, with ommochromes in turn repressing porphyrin biosynthesis. In this regard, it is interesting to speculate that kynurenine or other tryptophan metabolites, rather than ommochromes themselves, might have signaling activities that impact porphyrin levels.

Ommochromes and metalloporphyrins have diverse biological functions extending beyond pigmentation per se. Adaptive roles for porphyrins in the absence of metal coordination, while suggested by the 'physiological porphyria' in organisms such as fox squirrels, are not understood. Thus, we can only speculate as to possible body pigment function(s) in planarians. One possibility is

photoreception. *S. mediterranea* displays strong negative phototaxis. This is partly dependent on the melanin-producing pigment cups of the photoreceptors, yet animals lacking eye pigment retain photophobic behavior (*Lambrus et al., 2015*), and developing embryos display light avoidance prior to eye differentiation (*Sánchez Alvarado, 2003*). These observations point to the existence of one or more extraocular photoreceptors (*Paskin et al., 2014*). The burning pain experienced by some porphyria patients upon sunlight exposure (*Balwani and Desnick, 2012*) makes porphyrins logical candidates for further investigation.

## An animal model of acute porphyrias

Planarians have long attracted the interest of biologists investigating regeneration (*Newmark and Sánchez Alvarado, 2002*), and have recently emerged as a useful invertebrate model for human disorders including Usher syndrome (*Lapan and Reddien, 2012*) and cystic kidney disease (*Thi-Kim Vu et al., 2015*). Our results add acute porphyrias to this growing list.

The porphyrias are classified based on multiple criteria, including: 1) whether intermediates in heme biosynthesis are formed predominantly in the liver or bone marrow; 2) whether symptoms manifest primarily as skin lesions or neurovisceral episodes; and 3) the step at which heme biosynthesis is disrupted. Light-induced depigmentation in planarians combines aspects of multiple disease variants (*Figure 8*). UROS deficiency, the blockage in heme biosynthesis underlying CEP, is also the apparent determinant of physiological porphyrin biosynthesis in *S. mediterranea*. In the absence of this enzyme, its linear tetrapyrrole substrate hydroxymethylbilane undergoes spontaneous cyclization to form uroporphyrinogen I, a pathogenic isomer of uroporphyrinogen III that can in turn undergo spontaneous oxidation to form porphyrins. As in the human disease, our results implicate this biochemistry as the cause of severe cutaneous photosensitivity, but this is typically a chronic symptom in CEP, whereas it is induced by starvation in planarians. In the latter respect, *S. mediterranea* represents a unique animal model of acute porphyrias.

The episodic nature of acute porphyria symptoms is attributed to hepatic induction of *ALAS1*, 1 of 2 human isoforms of the gene, by endogenous and exogenous factors including dieting or fasting (*Balwani and Desnick, 2012*; *Karim et al., 2015*). This can pose a weight-loss challenge for individuals carrying disease alleles and may be a complicating factor in bariatric surgery (*Lopes et al., 2008*). *ALAS* and porphyrin levels are also increased in response to starvation in planarians (*Figure 7C,D*). These results are formally consistent with an increase in pigment cell number relative to other cell types as animal size decreases, but we regard this as unlikely because planarians become lighter in color, not darker, during prolonged starvation (*Newmark and Sánchez Alvarado, 2002*; *Miller and Newmark, 2012*). Furthermore, an increase in pigment cell density might be predicted to slow the rate of light-induced depigmentation, rather than accelerate it, and the rapid reversal of porphyrin-dependent photosensitivity when starved animals are re-fed (*Figure 7B,E*) is more consistent with a direct metabolic effect than a change in pigment cell density. In mice, fasting-induced *ALAS* upregulation is dependent on PGC-1α, and inhibited by insulin signaling (*Scassa et al., 2004*; *Handschin et al., 2005*). While we have been unable to identify a PGC-1α homolog in *S. mediterranea*, insulin-like signaling has been described (*Miller and Newmark, 2012*). Future experiments, including RNAi knockdown of nutrient-sensing genes and the development of a defined diet, will be important for identifying the mechanisms linking metabolic cues to *ALAS* expression in planarians.

Existing animal models of acute porphyrias commonly rely on chemical treatments to evoke phenotypes resembling human disease symptoms. The planarian model reported here complements these systems in several respects. First, off-target effects of porphyrogenic compounds can be avoided. Second, induction of photosensitization via starvation has the potential to reveal novel metabolic inputs into heme biosynthesis, as well as the toxic effects of pathway intermediates. Third, the fluorescence of fixed animals and easily inducible depigmentation response provide a convenient means of assessing porphyrin levels and their photosensitizing activity in vivo. This could prove especially useful in screening for drugs capable of reducing porphyrin biosynthesis (constitutive or induced) or alleviating porphyrin-based photosensitivity. The protective effect of DMTU against light exposure (*Figure 5B*) constitutes proof of principle for such an approach. Although porphyrias are usually manageable diseases, reliance on intravenous heme or liver transplantation to treat severe cases can result in significant complications (*Seth et al., 2007*). There is no approved prophylactic treatment for patients who suffer recurrent attacks. We suggest planarians represent an

experimentally tractable animal model in which to explore both physiological and pathological roles of porphyrins, and to seek new avenues for modulating their biosynthesis toward therapeutic benefit.

## Materials and methods

### Planarian maintenance

Asexual clonal populations of *S. mediterranea* (strain CIW4), *G. dorotocephala*, and *D. japonica* were maintained under standard laboratory conditions, as previously described (*Oviedo et al., 2008*). Except where otherwise indicated, animals were fed homogenized calf liver 1–3 times per week and starved for 7 to 8 days prior to use in experiments. Designated animals in *Figure 7E* were fed boiled egg yolk from organic, free-range chickens, supplemented with 10% (w/w) hemin chloride (Calbiochem) as indicated.

### Light exposure

Animals were exposed to white light from a compact fluorescent bulb (14W, 2700K, EcoSmart; see *Figure 1—figure supplement 3* for spectrum) or red light from a light-emitting diode (LED; 625 nm, BML Horticulture) in 12-well plates, 1 animal per well. Continuous red light exposure typically resulted in full depigmentation of 1 week-starved *S. mediterranea* within 3–5 days (smaller animals required less time on average). Continuous white light exposure resulted in 100% lethality prior to full depigmentation with a lamp height corresponding to 5,000 lux; lower intensities were less effective in inducing depigmentation. We empirically determined that exposure periods of 24, 48, 48, and 48 hr, interspersed with 24-hr recovery periods in a dark, 20°C incubator (standard laboratory conditions), almost completely eliminated lethality while achieving full depigmentation. This 10-day, intermittent exposure regimen was used in all white light experiments, unless otherwise indicated. UV-blocking glass (Tru Vue) was placed between animals and the light source to guard against any possible cell damage from sustained exposure to trace amounts of near-visible UVA radiation. KG3 IR/UVB-blocking glass (Omega Optical) was placed directly above sunlight-exposed animals for the experiment in *Figure 1—figure supplement 2A*.

### Microscopy and image acquisition

Photographs of whole animals were obtained with an Olympus SZX16 microscope equipped with a DP72 digital camera. Photographs of tissue sections (*Figure 4C*) and FISH (*Figure 4D*) were obtained with an Olympus BX53 microscope using the same camera. dFISH results (*Figure 4E*) were photographed with a Quorum Spinning Disk Confocal 2 (Olympus IX81 microscope and Hamamatsu C9100-13 EM-CCD camera). Raw images were captured using Olympus CellSens (SZX16 and BX53) or Perkin Elmer Velocity (confocal) software.

### Image processing and analysis

All photographs of live, light-exposed animals were acquired using identical camera settings and were processed only by placing images from different timepoints, different experimental conditions, etc. on a uniform, black background. Some variability in apparent animal color/pigmentation due to the position of the animal relative to the objective at the time of photographing, and position of the gooseneck lamps used for illumination was unavoidable. However, this variability was inconsequential compared to the pigmentation changes documented in the representative images shown in figures.

Linear adjustments (brightness and contrast) were made for images of animals labeled by WISH, FISH, dFISH, and TUNEL, or fixed to reveal porphyrin fluorescence, in order to best represent actual results. These adjustments were identical within a given experiment where comparisons were drawn between conditions.

### Ommochrome biochemistry

The absorption spectra in *Figures 2A* and *Figure 4—figure supplement 2B* were attained for a black *S. mediterranea* body pigment co-purifying with RNA (pigments like melanin commonly co-purify with nucleic acids in standard extraction procedures; *Giambernardi et al., 1998*). Briefly,

experimental and control animals were decapitated to remove the melanin-producing eye cups, and equal amounts of remaining trunk/tail fragments (0.1 g and 0.05 g wet tissue weight per sample in *Figures 2A* and *Figure 4—figure supplement 2B*, respectively) were homogenized in TRIzol reagent (ThermoFisher Scientific). Following chloroform extraction, the aqueous phase of each extract was precipitated with isopropanol. The black precipitate (not visible in the mock extraction from depigmented animals or *PBGD-1(RNAi)* extracts) was washed in ethanol, air dried, and resuspended in molecular biology grade water. Samples were treated with RNase I (NEB) for 1 hr at 37°C, after which TRIzol/chloroform extraction was repeated and the pigment and mock pigment samples were again resuspended in water. Degradation of RNA was confirmed by gel electrophoresis and absorption spectra were obtained on an HT Synergy plate reader (BioTek).

## Porphyrin biochemistry

We tested a variety of acid extraction protocols for porphyrin purification. All resulted in intense fluorescence under black light and absorption spectra with a pronounced Soret peak for *G. dorotocephala* and *D. japonica* extracts; in contrast, we typically detected minimal fluorescence for *S. mediterranea* extracts and/or were unable to resolve a clear Soret band. Homogenization in 1M sulfuric acid produced visible fluorescence and a small, but reproducible Soret peak. It is possible this reflects some degree of oxidation of non-fluorescent porphyrinogens to their oxidized, fluorescent form, yet addition of reagents commonly used for this purpose, including hydrogen peroxide and iodine or Lugol's solution (*Martásek et al., 1982*), had no obvious effect. Light exposure, which can photooxidize porphyrinogens, similarly failed to alter *S. mediterranea* fluorescence, either in extracts or in whole animals. *KMO-1* knockdown consistently enhanced extract fluorescence and resulted in a more prominent Soret peak, regardless of extraction method.

Results in *Figures 3C,D* and *7D* were obtained by centrifuging whole-animal $H_2SO_4$ homogenates for 10 min at 1452 x g, prior to analysis of absorbance/fluorescence. Equal amounts of tissue (30 mg wet weight) were used for each species/RNAi condition in *Figure 3C*. Absorption spectra were determined using an HT Synergy plate reader (BioTek). Fluorescence measurements (*Figure 7D*) were obtained using the same protocol and plate reader, with ~25 mg (wet tissue weight) *D. japonica* per sample. Readings were obtained with 400/30 nm (excitation) and 600/40 nm (emission) filters, and normalized for tissue weight. Porphyrin fluorescence was illustrated in *Figure 3C* by illuminating extracts in cuvettes with a black LED (400 nm, BML Horticulture).

## Whole-mount porphyrin fluorescence

Animals in *Figures 3A,B*, *6A*, and *Figure 6—figure supplement 1* were flash-killed in 5% N-acetyl cysteine (NAC) in PBS, transferred to 95% ethanol (*MacRae, 1961*), mounted on glass slides in this solution, and photographed with an Olympus SZX16 microscope using a 400–440 nm excitation filter.

## Chemical and hypoxia treatments

Animals were exposed to TDO inhibitor 680C91 (Focus Biomolecules), DMTU, and Ascorbic Acid (Sigma-Aldrich) as indicated in the relevant figure legends (*Figures 2E*, *5B*, and *Figure 5—figure supplement 1A*). Hypoxia (*Figure 5—figure supplement 1B*) was induced by placing animals in glass vials sealed with rubber stoppers and bubbling $N_2$ into the water (no air was bubbled into control vials). Dissolved $O_2$ was reduced by approximately 85% using this approach. Animals were immediately subjected to a 5-hr pulse of red LED exposure (in glass vials) and then returned to normoxic conditions overnight (sustained hypoxia was lethal). This process was repeated a total of 4 times over 5 days (days 0, 1, 3, and 4), and depigmentation was assessed 3 days after the final light exposure.

## Reciprocal BLAST and cloning

To identify candidate ommochrome and porphyrin biosynthesis enzymes (*Figures 2B* and *4A*), human and mouse protein sequences were used as queries in TBLASTN searches against *S. mediterranea* genomic and EST databases (*Labbé et al., 2012*; *Robb et al., 2015*; *Zhu et al., 2015*). Top hits were used as queries in reciprocal BLASTX searches against the non-redundant protein database (NCBI), and discarded if results did not match the identity of the original query. cDNA

sequences were then cloned into a double-stranded RNA (dsRNA) expression vector (pT4P; *Rink et al., 2009*) by RT-PCR, using primers shown in *Supplementary file 1*. We were unable to amplify *ALAD-2*, *UROD-2*, and *FECH-2* by RT-PCR; there were also no ESTs matching these genomic ORFs in the *S. mediterranea* Genome Database (*Robb et al., 2015*), raising the possibility they are pseudogenes.

## Whole-mount in situ hybridization and TUNEL

WISH, FISH, and dFISH were performed as previously described (*Pearson et al., 2009*; *Zhu et al., 2015*; *Currie et al., 2016*), using riboprobes prepared from pT4P clones, and imaged as described above. Whole-mount TUNEL was also performed according to published protocols (*Pellettieri et al., 2010*), and quantified using ImageJ software (http://rsb.info.nih.gov/ij). Designated animals in *Figure 7—figure supplement 1* were exposed to 1250 rad of gamma radiation, a sublethal dose that depletes cycling stem cells (*Wagner et al., 2011*), using a Cs-137 source, 81-14R irradiator (J.L. Shepherd & Associates). Irradiation was completed 24 hr prior to fixation.

## RNA interference

dsRNA-expressing *E. coli* cultures were prepared using pT4P clones, mixed with homogenized calf liver, and fed to animals as previously described (*Zhu et al., 2015*). An RNAi vector with *C. elegans unc-22* was used as a negative control. To evaluate pigmentation changes in intact animals as a result of existing pigment turnover (*Figures 2D* and *4F*, top panels), animals were fed RNAi food 12 times over 3.5 weeks (days 0, 2, 4, 7, 9, 11, 14, 16, 18, 21, 23, and 25), discarding animals that did not eat after each feeding. Photographs were taken 3 days after the final feeding (day 28). For genes that generated phenotypes unrelated to pigmentation (*Figure 4—figure supplement 1A*), feedings were discontinued when less than 50% of animals ate. The experiment in *Figure 6B* followed the same RNAi feeding schedule (animals were light exposed 7 days after the final RNAi feeding). To evaluate pigmentation changes arising from defects in pigment biosynthesis during regeneration of new, initially unpigmented tissue (*Figures 2D* and *4F*, bottom panels), animals were amputated to form head, trunk, and tail fragments on day 28. Regenerating trunk fragments were fed again on days 42, 44, and 46, and anterior blastemas were photographed on day 49, at 21 days post-amputation. In cases involving the potential for genetic redundancy (e.g., *KFM-1* and *-2*), equal amounts of dsRNA-expressing bacterial cultures were combined to prepare RNAi food (*Gurley et al., 2008*).

All other RNAi experiments involved 4 RNAi feedings (days 0, 2, 4, and 7), followed by a single round of amputation on day 8 to promote tissue turnover, and another 4 RNAi feedings on the same schedule initiated 10 to 12 days post-amputation. Phenotypes were analyzed 7 days after the final RNAi feeding. Animals were rinsed and observed every 1–2 days.

## qRT-PCR

*ALAS* qRT-PCR was performed in triplicate as previously described (*Lin and Pearson, 2014*), using size-matched animals fed 4 times in 1 week with dyed calf liver at the start of the experiment. Animals were starved for 24 hr or 14 days, as indicated in *Figure 7C*, prior to preparation and reverse transcription of total RNA using TRIzol reagent (ThermoFisher Scientific) and SuperScript III reverse transcriptase (Invitrogen). cDNA was amplified with LightCycler 480 SYBR Green I Master reaction mix (Roche) in a CFX96 Touch Real-Time PCR Detection System (Bio-Rad), and results were normalized using the ubiquitously expressed *GAPDH* gene as a reference. Primers were designed to span exon-exon boundaries:
*ALAS*: Forward - CAACGAGTGATTGTTAAGTCTGG; Reverse - GACAGACATTCATTTGGTTGCTC
*GAPDH*: Forward - AGCTCCATTGGCGAAAGTTA; Reverse - CTTTTGCTGCACCAGTTGAA

## Acknowledgements

We thank Casey Kimball, Haley Zanga, Vanessa Poirier, Colby Easter, and Kim Tu for assistance with completion of experiments, and Paul Baures, David Mullins, Steven Fiering, Charles Cole, Mauricio da Silva Baptista, and staff scientists at Omega Optical for support and advice. We also appreciate the curiosity of students in Keene State College's Stem Cells and Regeneration class that led to the discovery of the light-induced depigmentation response.

## Additional information

### Funding

| Funder | Grant reference number | Author |
|---|---|---|
| Canadian Institutes of Health Research | MOP-130294 | Xinwen He |
| Ontario Institute for Cancer Research | IA-026 | Bret J Pearson |
| National Institutes of Health | P20GM103506 | Jason Pellettieri |
| National Institutes of Health | 1R15GM107826-01 | Jason Pellettieri |
| National Science Foundation | IOS-1445541 | Jason Pellettieri |

The funders had no role in study design, data collection and interpretation, or the decision to submit the work for publication. The content of this article is solely the responsibility of the authors and does not necessarily represent the official views of the funders.

### Author contributions

BMS, JPD, ERN, MSB, LEN, EB-M, XH, Conception and design, Acquisition of data, Analysis and interpretation of data; BJP, Conception and design, Analysis and interpretation of data; JP, Conception and design, Analysis and interpretation of data, Drafting or revising the article

### Author ORCIDs

Jason Pellettieri, http://orcid.org/0000-0002-7800-4904

## Additional files

### Supplementary files

• Supplementary file 1. Cloning of *S. mediterranea* ommochrome and porphyrin/heme biosynthesis genes. See Materials and methods for details. [1]Smed Unigene transcripts are available at the *Schmidtea mediterranea* Genome Database (*Robb et al., 2015*). [2]Smed_ASXL transcripts are available under NCBI BioProject PRJNA215411. [3]Not cloned (no RT-PCR product).

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
