## [Decision Letter]

Thank you for submitting your work entitled "Light-induced depigmentation in planarians models the pathophysiology of acute porphyrias" for consideration by *eLife*. Your article has been reviewed by two peer reviewers, and the evaluation has been overseen by Alejandro Sánchez Alvarado as the Reviewing Editor and Diethard Tautz as the Senior Editor.

The reviewers have discussed the reviews with one another and the Reviewing Editor has drafted this decision to help you prepare a revised submission

Summary:

This very interesting manuscript describes remarkable parallels between planarian photosensitivity and that observed in human porphyrias. The authors discovered a striking photosensitivity of planarian body pigmentation, namely that prolonged light exposure results in depigmentation in planarians. Equally remarkable is the fact that this discovery was made by undergraduates during the course of a general biology class. Following up on this observation, the authors show that this process is not simple photobleaching, but rather is caused by active elimination of pigment cells. Through the systematic analysis of ommochrome and porphyrin biosynthetic pathways, the authors identified several genes required for producing planarian pigment. RNAi knockdown of *KMO-1*, an enzyme involved in ommochrome synthesis that is expressed in pigment cells, leads to alterations in body color and increased porphyrin levels. The authors report that genes encoding heme synthetic enzymes (that generate porphyrin intermediates) are expressed in pigment cells, and the first three enzymes in the heme synthetic pathway are also required for proper pigmentation. To explore the potential role of porphyrins in light-induced depigmentation, they knock down genes involved in the heme/porphyrin biosynthesis pathway and show that knockdown of one gene, *PBGD-1*, eliminates light-induced depigmentation, and suppresses the heightened photosensitivity of *KMO-1(RNAi)* animals. The authors also report that the light-induced depigmentation is observed in starved, but not recently fed, animals. This work is carefully performed and thoughtfully presented. It lays the foundation for subsequent studies to investigate the biology of pigment cell differentiation and chemical identification of the pigments themselves (both of which I believe are beyond the scope of this study). The development of methods to remove pigmentation from planarians could have significant impacts on tool development in that field, and the connection to human porphyrias makes a compelling case for additional studies of this system. Altogether, we anticipate that the work will be of interest to the broad readership of this journal.

Essential revisions:

Before the paper can be further considered for publication, the reviewers, after consulting with each other, would like to see the authors address the following issues:

1) It is important for the authors to provide additional evidence for the proposed parallel between the consequences of decreased nutrient intake in planarians and humans. Because planarians are fed calf liver, which is presumably rich in heme, they could be looking at the dietary intake of heme and its specific consequences on the porphyrin biosynthetic pathway and not a generic response to feeding. Heme synthesis is known to be feedback-inhibited by heme, e.g. ALAS downregulation by heme (PMID: 10522552). Is it possible for the authors to use alternative food sources that may contain less heme (i.e., egg yolk) that would allow them to more clearly distinguish these options? Are ALAS transcript levels affected by starvation and/or different diets?

2) The author's model (subsection “Ommochrome and porphyrin biosynthesis in flatworms”, third paragraph and Figure 8) suggests that ommochromes negatively regulate porphyrin levels based on the observation that *KMO-1(RNAi)* causes elevation of porphyrins. However, I think another possibility worth considering is that derivatives of ommochromes and porphyrins (or HMB) could be used jointly as precursors for a downstream biosynthetic step, potentially to produce the final brown pigment itself. This model would account for why *KMO-1(RNAi)*, a treatment expected to reduce/eliminate ommochrome synthesis, results in accumulation of a porphyrin. Another possibility in line with author's current model could be that some hormonal/signaling action of tryptophan derivative can suppress porphyrin synthesis rather than ommochromes themselves having some interaction with porphyrins.

3) It is rather interesting that starvation accentuates the photosensitivity of pigmentation. Is this perhaps due in part to a general difference in cell death responsiveness due to nutrient status in planarians? Additionally, does starvation increase porphyrin abundance (as measured by red fluorescence) or impact the response to normal levels of porphyrins? These experiments would clarify the mechanism of how nutrient intake modulates light sensitivity.

---

## [Author Response]

Essential revisions:

Before the paper can be further considered for publication, the reviewers, after consulting with each other, would like to see the authors address the following issues: 1) It is important for the authors to provide additional evidence for the proposed parallel between the consequences of decreased nutrient intake in planarians and humans. Because planarians are fed calf liver, which is presumably rich in heme, they could be looking at the dietary intake of heme and its specific consequences on the porphyrin biosynthetic pathway and not a generic response to feeding. Heme synthesis is known to be feedback-inhibited by heme, e.g. ALAS downregulation by heme (PMID: 10522552). Is it possible for the authors to use alternative food sources that may contain less heme (i.e., egg yolk) that would allow them to more clearly distinguish these options? Are ALAS transcript levels affected by starvation and/or different diets?

Given the feedback inhibition in the heme biosynthesis pathway (cited in our original manuscript), we agree that comparing the photoprotective effects of different laboratory diets is a useful experiment. Our revised manuscript shows that egg yolk is slightly less effective than calf liver in reversing starvation-induced photosensitivity (Figure 7), as might be expected if liver suppresses porphyrin biosynthesis merely because of its relatively high heme content. However, addition of exogenous heme to egg yolk had no discernable effect. We also show that both *ALAS* expression and porphyrin content increase with starvation (using qRT-PCR and measurements of porphyrin fluorescence in tissue homogenates – see Figure 7, respectively), as occurs in mouse models of acute porphyrias (e.g., Handschin et al., 2005). While these new results do not exclude a role for dietary heme in regulating porphyrin biosynthesis in planarians, they do provide further support for the existence of other metabolic control mechanisms, underscoring planarians’ value as an acute porphyria disease model. Further experiments (e.g., developing a defined diet) will be necessary to characterize the molecular mechanisms linking nutrient intake to *ALAS* induction, but we respectfully suggest that work is beyond the scope of the present manuscript. We have emphasized the need for such future experiments in the revised Discussion section (”An animal model of acute porphyrias”, third paragraph).

2) The author's model (subsection “Ommochrome and porphyrin biosynthesis in flatworms”, third paragraph and Figure 8) suggests that ommochromes negatively regulate porphyrin levels based on the observation that KMO-1(RNAi) causes elevation of porphyrins. However, I think another possibility worth considering is that derivatives of ommochromes and porphyrins (or HMB) could be used jointly as precursors for a downstream biosynthetic step, potentially to produce the final brown pigment itself. This model would account for why KMO-1(RNAi), a treatment expected to reduce/eliminate ommochrome synthesis, results in accumulation of a porphyrin. Another possibility in line with author's current model could be that some hormonal/signaling action of tryptophan derivative can suppress porphyrin synthesis rather than ommochromes themselves having some interaction with porphyrins.

We agree that these possibilities are both consistent with our results and worth addressing. The potential for joint use of ommochromes and porphyrins in a downstream biosynthetic step is explicitly addressed in our revised manuscript in the Figure 8 legend, as well as in the Discussion: “One possible explanation of these results is that ommochromes and porphyrins are jointly used as precursors in a downstream biosynthetic step to generate a single, brown body pigment…”. The possibility that tryptophan derivatives have signaling roles that impact porphyrin biosynthesis is addressed in the Discussion: “…it is interesting to speculate that kynurenine or other tryptophan metabolites, rather than ommochromes themselves, might have signaling activities that impact porphyrin levels.” Along with these additions to the text, we have also removed the arrows denoting possible regulatory interactions between ommochromes and porphyrins in the model (Figure 8) to avoid giving the impression that we favor this scenario over the above (and other) equally plausible alternatives.

3) It is rather interesting that starvation accentuates the photosensitivity of pigmentation. Is this perhaps due in part to a general difference in cell death responsiveness due to nutrient status in planarians? Additionally, does starvation increase porphyrin abundance (as measured by red fluorescence) or impact the response to normal levels of porphyrins? These experiments would clarify the mechanism of how nutrient intake modulates light sensitivity.

We tested whether starvation leads to nonspecific cell death responsiveness by comparing TUNEL staining in 7 day and 14 day-starved animals exposed to a sublethal dose of gamma radiation. As shown in Figure 7—figure supplement 1, there was no significant difference between these conditions. In contrast, we found that there is a small, but clearly significant increase in porphyrin levels in *D. japonica* (used because porphyrin fluorescence is much higher than in *S. mediterranea* and therefore more easily quantified in tissue homogenates) from 7 to 14 days post-feeding (Figure 7); a further increase was evident at 28 days post-feeding. Together with the increase in *ALAS* expression observed in starved animals (see above), this not only clarifies how nutrient intake modulates photosensitivity, but adds to the parallels between light-induced depigmentation in planarians and the effects of dieting or fasting observed in acute porphyrias.